# Potential Neuromodulation of the Cardio-Renal Syndrome

**DOI:** 10.3390/jcm12030803

**Published:** 2023-01-19

**Authors:** Irving H. Zucker, Zhiqiu Xia, Han-Jun Wang

**Affiliations:** 1Department of Cellular and Integrative Physiology, University of Nebraska Medical Center, Omaha, NE 68198, USA; 2Department of Anesthesiology, University of Nebraska Medical Center, Omaha, NE 68198, USA

**Keywords:** heart failure, sympathetic nerve activity, renal function, cardiac reflexes, denervation

## Abstract

The cardio-renal syndrome (CRS) type 2 is defined as a progressive loss of renal function following a primary insult to the myocardium that may be either acute or chronic but is accompanied by a decline in myocardial pump performance. The treatment of patients with CRS is difficult, and the disease often progresses to end-stage renal disease that is refractory to conventional therapy. While a good deal of information is known concerning renal injury in the CRS, less is understood about how reflex control of renal sympathetic nerve activity affects this syndrome. In this review, we provide insight into the role of the renal nerves, both from the afferent or sensory side and from the efferent side, in mediating renal dysfunction in CRS. We discuss how interventions such as renal denervation and abrogation of systemic reflexes may be used to alleviate renal dysfunction in the setting of chronic heart failure. We specifically focus on a novel cardiac sensory reflex that is sensitized in heart failure and activates the sympathetic nervous system, especially outflow to the kidney. This so-called Cardiac Sympathetic Afferent Reflex (CSAR) can be ablated using the potent neurotoxin resinferitoxin due to the high expression of Transient Receptor Potential Vanilloid 1 (TRPV1) receptors. Following ablation of the CSAR, several markers of renal dysfunction are reversed in the post-myocardial infarction heart failure state. This review puts forth the novel idea of neuromodulation at the cardiac level in the treatment of CRS Type 2.

## 1. Introduction

Chronic heart failure (CHF) is a growing epidemic in the Western world, with an estimated 6 million cases in the United States alone [1]. This increase is mediated by a multitude of factors, including an increase in the age of the population, obesity, inactivity, coronary arterial disease, and a scattering of other genetic and environmental factors. The syndrome of CHF impacts every organ system. Ultimately, multi-organ failure ensues, resulting in increased morbidity and mortality. The kidney is especially vulnerable to ischemia in the setting of CHF. Even with optimal therapy, a subset of patients develops chronic renal failure referred to as Cardiorenal Syndrome Type 2 (CRS 2), which results in a rapidly deteriorating scenario that involves both under-perfusion of the kidney and tubular damage with a grave prognosis [2].

The term CRS implies acute or chronic injury to the heart and kidneys that often involves a temporal sequence of disease initiation and progression. The classification of CRS is generally divided into five subtypes. Types 1 and 2 involve acute and chronic cardiovascular disease (CVD) scenarios leading to acute kidney injury (AKI) or accelerated chronic kidney disease (CKD). Types 3 and 4 describe AKI and CKD, respectively, leading primarily to heart failure. Finally, CRS type 5 describes a systemic insult to both heart and the kidneys, such as sepsis, where both organs are injured simultaneously in patients with previously normal heart and kidney function at baseline. The hemodynamic mechanisms by which renal function deteriorates in CHF are multiple and have been well described in the literature [3]. The traditional dogma of how CRS 2 develops in heart failure is that it results, in a large degree, from heart failure-induced hypoperfusion of the kidneys. Neurohumoral activation of the renin–angiotensin–aldosterone system (RAAS) plays a major role in the pathogenesis of CRS [4]. Inflammation and oxidative stress in CHF also perpetuate renal damage [5,6]. In addition, renal dysfunction in heart failure is now also thought to develop as a result of increased cardiac congestion, increased right atrial pressure, and increased central venous pressure (CVP), which results in the development of renal venous hypertension, renal venous congestion, increased renal fibrogenesis, and eventually the loss of renal function. Renal venous congestion and loss of renal function in combination with (neuro)hormonal responses and potentially altered intra-abdominal and/or intrasplanchnic pressures which may contribute to fluid overload and increased venous return in a vicious cycle. Medical therapy is divided into several classes of agents that have favorable effects on both the heart and the kidneys in the setting of CRS management. In general, evidence strongly supports the use of angiotensin-converting enzyme inhibitors (ACEi) in both heart failure and renal disease. For those with ACEi intolerance, Ang II receptor antagonists (ARB) play an important therapeutic role. However, despite early enthusiasm for dual therapy, the use of ACEi and ARBs together has resulted in an unfavorable risk/benefit ratio with higher rates of acute kidney injury, hyperkalemia, and hypotension [7]. Considerable effort has also been undertaken to understand the range of metabolic perturbations that occur in CKD and the evaluation of modifications for renal and cardiac outcomes. However, to date, modifications of a single factor such as hemoglobin, parathyroid hormone, or reduction in oxidative stress have resulted in no demonstrable benefit for outcomes in the CRS, leading to the cessation of clinical development for these agents [8,9,10]. In patients who are resistant to pharmacological interventions, there is a progression to hyponatremia and death [11]. Ultimately, drug-resistant patients are placed on hemodialysis [12].

There are several putative pathways that may be amenable to specific types of non-pharmacological interventions, as summarized in Figure 1. Several investigators have proposed interventions to increase renal blood flow in the face of low cardiac output using extracorporeal perfusion of the kidneys [13,14,15,16]. Similarly, a major mechanism that contributes to renal failure in CHF is the elevated venous pressures that are associated with the low cardiac output state in CHF. Certainly, a reduction in renal venous pressure would be beneficial in ameliorating the reduced renal blood flow, GFR, and high renal interstitial pressures. Therefore, it is critical to discover novel mechanisms underlying renal dysfunction in CHF and develop effective alternative therapies. This review will focus on several neuromodulation therapies to treat CRS 2.

Given the fact that our understanding of many of the mechanisms related to deteriorating renal function and CRS in heart failure have been clarified over the years, there is still no effective therapy. This issue requires a fresh look at other mechanisms that may be novel targets for intervention. 

## 2. Neural Regulation of Renal Function

The neural innervation of the kidney has intrigued anatomists, physiologists, and clinicians for many years [17,18,19,20]. Early work described the anatomy of the post-ganglionic innervation in detail [20,21,22]. Electron microscopic studies clearly show nerve terminals adjacent to many structures in the renal cortex and the pelvis [23]. The renal vascular and tubular elements have all been described to have adjacent nerve terminals [20]. Evidence for both afferent and efferent endings has been demonstrated using both neuropeptide staining and anatomical landmarks [24]. The efferent innervation is primarily of sympathetic origin, targeting the tubular system and the intrarenal vasculature with a supply of norepinephrine (NE), neuropeptide Y (NPY), and ATP, whose release can be modulated rapidly [25,26,27]. While there is no parasympathetic innervation of the kidney per se, acetylcholine esterase has been identified in the kidney [28,29]. The sympathetic innervation of the kidney is a powerful regulator of vascular tone, sodium reabsorption, oxidative stress, and inflammation [30,31]. Activation of the renal sympathetic nerves can reduce renal blood flow to near zero, especially in cardiovascular disease such as CHF [32]. 

Renal sensory endings have also been well described, especially at the renal hilus and in the minor and major calyces [33]. These afferents release classical neuropeptides such as calcitonin gene-related peptide (CGRP) and Substance P [33]. They respond to a variety of stimuli, including mechanical deformation, renal interstitial pressure, osmolality, pH changes, and pro-inflammatory agents [33]. They have been shown to participate in renal reflexes that regulate both sympathetic outflow, renin release, and vasopressin secretion [34]. Classical pain afferents in the kidney respond to algesic agents such as bradykinin [35,36,37]. Renal afferents project to several areas in the central nervous system to regulate sympathetic outflow, renin release, and vasopressin release. This is especially true in the paraventricular nucleus (PVN) of the hypothalamus. In this regard, renal afferents are sensitized in the setting of CHF [38,39,40]. Interestingly, the activation of renal efferent pathways may be mediated by inflammatory products and initiate neuroinflammation in the kidney, thus contributing to renal dysfunction [41,42]. It is not clear, however, if activation of renal afferents translates into chronic effects on renal function in the setting of CHF.

## 3. Impaired Cardiovascular Reflexes in CHF

A large body of literature solidifies the concept that sympatho-inhibitory cardiovascular reflexes (e.g., arterial baroreflex, vagal reflexes) are blunted in CHF, while sympatho-excitatory reflexes (e.g., arterial chemoreflex, cardiac sympathetic afferent reflex, muscle metaboreflex; reno-renal reflex) are sensitized [31,43,44,45]. Altered cardiovascular reflex function in CHF is undoubtedly mediated by multiple mechanisms, including changes in afferent sensory function, modulation of central processing and neurotransmitter release and receptor expression and binding, as well as augmentation of several humoral substances that may contribute to sympatho-excitation such as Angiotensin II and endothelin-1 [46]. While global sympathetic outflow is augmented in heart failure, the impact on the kidney has major consequences, given that approximately 20% of the cardiac output perfuses the kidneys. Thus, the idea that renal denervation is efficacious in CHF has been put forward. Figure 2 shows the role of the major inhibitory and excitatory cardiovascular reflexes on sympathetic outflow.

## 4. Renal Denervation

It has long been known that interruption of the renal sympathetic nerves results in short-term renal vasodilation, an increase in renal blood flow, and a reduction in renin release [47,48]. However, as is the case for renal transplantation, renal blood flow tends to be normal under most circumstances following chronic renal denervation [49]. Renal denervation, as a therapy for drug-resistant hypertension in humans, was introduced in 2009 using a radiofrequency catheter ablation technique [50]. This proof of principle study led to a plethora of clinical trials and animal studies that have evaluated its safety and efficacy [41,51,52,53]. Renal denervation is now used to treat severe, drug-resistant hypertension as well as other diseases, including heart failure [54,55]. While it now seems clear that renal denervation is efficacious in both hypertension and some forms of CHF (both HFpEF and HFrEF) [56,57], the mechanisms involved are still enigmatic. It is likely that both afferent and efferent pathways play a critical role. However, there is little question that augmented renal sympathetic release of NE promotes increased renin release, oxidative stress, and inflammation in the kidney [58], contributing to renal dysfunction, volume expansion, and worsening heart failure. In CHF, renal sympathetic innervation evokes increased vasomotion that is reversed by renal denervation [59]. It is also likely that sympathetically mediated venous congestion results in a decrease in salt and water reabsorption in CHF and that adrenergic and angiotensinergic targeting of renal epithelia promote salt and water reabsorption [60].

## 5. Spinal Cord Stimulation

Spinal cord stimulation has been used to block pain for many years. High-frequency stimulation can also inhibit sympathetic outflow to the splanchnic area [61,62,63]. Unfortunately, spinal cord stimulation in patients with heart failure did not show any benefit [64].

## 6. Carotid Baroreflex Activation

The arterial baroreflex is a major sympatho-inhibitory reflex that controls short-term blood pressure variability. Animal and human studies have shown that the sensitivity of the arterial baroreflex is blunted in heart failure [65,66,67,68]. Building on early work designed to reduce sympathetic outflow to the heart in the post-myocardial infarction state [69,70], the idea of baroreceptor activation therapy was reintroduced to treat resistant hypertension [71]. Baroreflex activation therapy for heart failure soon followed [72]. Several clinical trials were positive and showed improved exercise tolerance and increased quality of life along with reduced plasma norepinephrine and decreased muscle sympathetic nerve activity [73,74,75,76].

In dogs with pacing-induced heart failure, Zucker et al. [77] showed enhanced survival following baroreflex activation therapy. Other canine studies also showed significant alterations in fluid balance and renin release during baroreflex activation [78]. However, there have been few studies that have examined the effects of BAT on renal function in heart failure that has progressed to the level of CRS Type 2.

It should also be noted that there have been attempts to treat CHF with carotid body ablation, given that the carotid chemoreflex is sensitized in heart failure [79,80,81,82] and drives increased sympathetic outflow. There does seem to be a positive effect of this intervention [83,84]; however, the focus has been on sympathetic tone and cardiac function rather than renal function *per se*.

## 7. Cardiac Afferent Denervation

Sensory input from the heart consists of vagal and spinal pathways that mediate inhibitory and excitatory reflexes, respectively. Previous data from our laboratory have clearly shown that vagal sensory discharge and cardiac mechano-reflexes (mostly from the atria) are blunted in CHF [66]. Because these tend to be sympatho-inhibitory, they ostensibly contribute to an increase in sympathetic outflow and increases in both renin and vasopressin release [85]. Direct electrophysiological recording from vagal sensory afferents indicates that while endings conducting via large-diameter myelinated fibers have reduced sensitivity, small-diameter afferents may have augmented sensitivity, especially in response to specific agonists [86,87]. On the other hand, cardiac afferents that directly enter the spinal cord through the thoracic dorsal root ganglia (DRG) consist of mostly small fibers with low conduction velocity (c-fibers) and are extremely sensitive to molecules released during or after coronary ischemia [88]. These afferents are sensitized in heart failure [43] and contribute to both the sensation of cardiac pain during ischemia [89] and to augmented sympatho-excitatory reflexes that can become chronic in the setting of heart failure. 

Our laboratory has consistently shown a sensitization of the so-called cardiac sympathetic afferent reflex (CSAR) in animals with heart failure [43,44,68]. Epicardial application of bradykinin or capsaicin results in the augmented discharge of cardiac spinal afferents and an increase in renal sympathetic nerve activity in animals with heart failure. It has been well documented that many of the cardiac spinal afferents exist on the epicardial surface and express the non-specific cation channel TRPV1 [44]. Importantly, the permanent destruction of TRPV1-containing afferents completely abolishes the response to bradykinin and capsaicin [44]. In this study, Wang et al. [44] have shown that abolition of the CSAR using an epicardial application of the potent TRPV1 agonist resiniferitoxin (RTX) at the time of myocardial infarction reduces sympathetic outflow to the heart and kidney, decreases markers of inflammation, restores diastolic dysfunction, and reduces cardiac fibrosis. Furthermore, this intervention significantly increases longevity [90]. These findings raise the possibility that cardiac afferent ablation may ameliorate CRS in heart failure.

In a recent study by Xia et al. [90] carried out in rats with MI-induced CHF, the role of cardiac spinal afferents in mediating and exacerbating CRS was investigated. In animals with severe heart failure (18 weeks post MI), BUN and creatinine were elevated, and GFR was reduced. Urine volume was reduced, and microalbuminuria was present. These parameters were all reversed in animals pretreated with RTX. In addition, cardiac fibrosis was reduced [44] along with histologically verified renal fibrosis [90]. Following RNA sequencing, the highest expressed gene in the renal cortex of vehicle-treated rats with CHF compared to sham rats was Havcr1 (~300 fold), the gene that codes for Kim1 (kidney injury marker). This was verified by the measurement of the Kim1 protein. Ngal also showed a significant increase in the CHF state. In the group of rats treated with RTX, there was a significant reduction in both Kim1 and Ngal. In addition, intrarenal vasoconstriction and the reduction in renal blood flow that accompanies epicardial bradykinin in MI vehicle-treated rats were ameliorated in rats pretreated with RTX. Chronic CSAR ablation also attenuated the increase in central venous pressure in CHF rats, a mechanism that may contribute to high renal venous pressure and contribute to a decrease in GFR (Figure 3). As indicated above, RTX treatment reduced mortality by approximately 40% (Figure 3). Two questions arise from this study. First, can one interrupt the CSAR by ablation of thoracic afferents that traverse the stellate ganglia? Second, does simple renal denervation achieve similar results as cardiac spinal afferent ablation?

When RTX was given into the stellate ganglia of CHF rats, there was a reduction in Kim1 compared to vehicle-treated animals. This was not observed in the renal medulla, however. In addition, unilateral renal denervation carried out at 4 weeks after MI improved cardiac function when measured at 18 weeks post-MI. Significantly, unilateral renal denervation increased GFR compared to heart failure rats subjected to a sham denervation procedure. It should be emphasized that none of the interventions above altered infarct size or ejection fraction. 

Thus, these data strongly suggest that the reduction of renal sympathetic nerve activity either by direct renal denervation or by interrupting specific cardiac spinal reflexes in the setting of CHF impacts renal function and renal injury. This technique could be potentially therapeutic in CRS Type 2 when other therapies are ineffective.

## 8. Conclusions

Recent literature and white papers advocate for more studies on neuromodulation for the treatment of both pain syndromes and cardiovascular disease [91]. This line of investigation includes both chemical modulation and electrical devices (e.g., spinal cord stimulation). Direct modulation of renal sympathetic nerve activity in the setting of cardiovascular disease characterized by high levels of sympathetic tone is likely to be beneficial in reducing renal dysfunction in heart failure. The novel innovation put forth here is that interruption of excitatory neural input from the diseased heart may also have a beneficial effect on renal function in heart failure. While innervations such as RTX application may be irreversible and interrupt thoracic pain pathways, we believe this is a minor concern in the setting of CRS Type 2, where cardiovascular collapse is imminent.

## Figures and Tables

**Figure 1 jcm-12-00803-f001:**
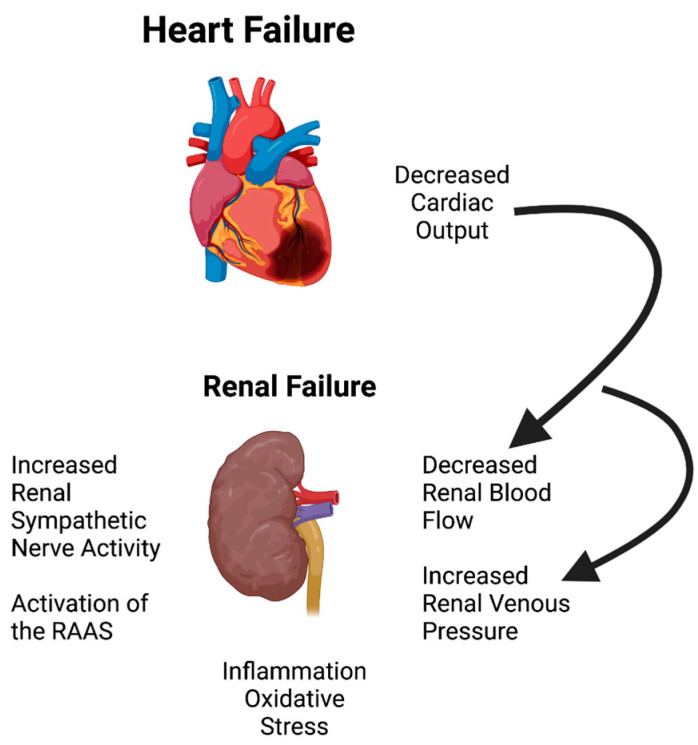
Renal dysfunction in heart failure is mediated by reduced perfusion, increased renal venous pressure, increased sympathetic outflow and activation of the RAAS, inflammatory pathways, and oxidative stress, all evoking cardio-renal syndrome Type 2.

**Figure 2 jcm-12-00803-f002:**
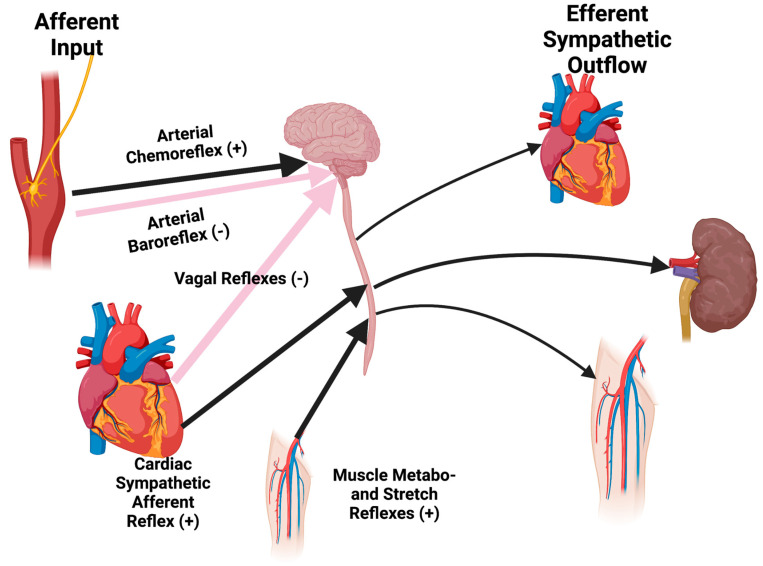
Heart failure is associated with abnormalities in both inhibitory and excitatory cardiovascular reflexes emanating from the periphery and the heart itself. These abnormal reflexes result in an increase in sympathetic outflow to the heart, peripheral vasculature, and kidney.

**Figure 3 jcm-12-00803-f003:**
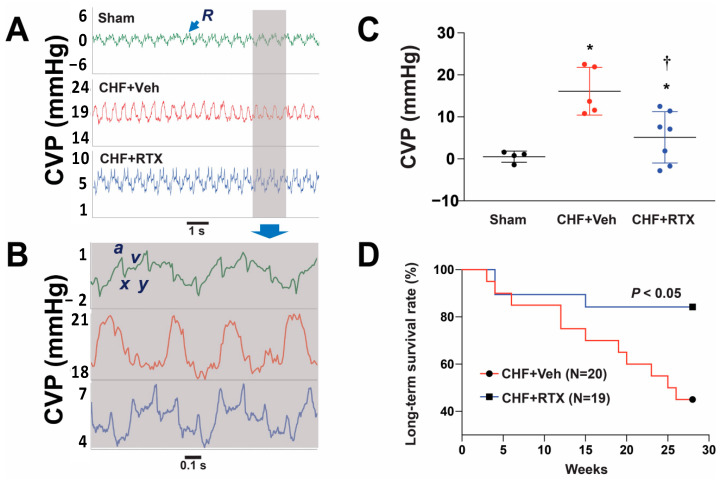
CSAR ablation by RTX reduced the increased central venous pressure (CVP) and improved survival in CHF rats. Original recording (**A**,**B**) and summary data (**C**) showing that cardiac spinal afferent ablation by RTX reduced the increased CVP in CHF rats. R: respiratory wave; a wave: atrial contraction; x descent: atrial relaxation; v wave: systolic filling of atrium; y descent: early ventricular filling. Sham group, n = 4; CHF + Veh group, n = 5; CHF + RTX group, n = 7. * *p* < 0.05 vs. Sham. †, *p* < 0.05 vs. CHF + Vehicle. (**D**) Kaplan–Meier survival rates between CHF rats treated with epicardial application of RTX (n = 19) or vehicle (n = 20) at the time of MI. Figure from [90], with permission.

## Data Availability

Please refer to original articles in this review for data archiving information.

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
