# Peer review of "Potential Neuromodulation of the Cardio-Renal Syndrome"

_jcm, 2023, doi:10.3390/jcm12030803_

Round 1
Reviewer 1 Report
The paper is characterized by a broad introduction on the epidemiology of chronic heart failure and on the classification of cardio-renal syndrome. There is a focus on the commonly considered pathophysiological aspects, where the leading issue is the criterion of renal hypoperfusion. In addition to the role of RAAS activation in the pathogenesis of cardiorenal syndrome, an important role is recognized for the Mullens' hypothesis, regarding the hemodynamic role characterized by the increase of right atrial pressure and central venous pressure, able to developing renal venous hypertension and congestion, up to the loss of renal function.
There are also the following paragraphs dedicated to neural regulation of renal function, and to impaired cardiovascular reflexes in CHF and renal denervation and finally to cardiac afferent denervation.
The paper and the introduction are presented in a not exactly convincing way with regard to epidemiological and therapeutic data, reserved for the population suffering from heart failure. Indeed, the subsequent analysis addresses pathophysiological concepts on neuroregulation of renal function and on inadequate cardiovascular reflexes in chronic heart failure. On the other hand, chapters 4 and 7 should represent those characterized by contents more in line with the new and more interesting evidence that the authors want to highlight; this first topic should, in my opinion, be better developed and deepened above all by looking for correlations and integrations with chapter 7.
It must also be emphasized that the experiences reported in chapter 7 are objectively subjects of experimentation on animals; the latter represents a model studied in absolute terms, without the context of an optimized side-by-side medical therapy.
Otherwise, the prospect of these therapeutic interventions in humans cannot ignore the presence of an optimized medical therapy of heart failure with various levels of inhibition of the RAAS.
In conclusion, in my opinion, it would be desirable to have a greater integration between the various topics covered, both in the context of neurohormonal interaction and strictly haemodynamic aspects. It should be defined in the most appropriate way
the clinical studies carried out on humans and animals for the various hypotheses considered.
Author Response
Thank you for these comments. This mini-review is meant to bring attention to the idea that neuromodulation may be an alternative strategy to treat the CRS-2. We agree that this notion is based primarily on pre-clinical data and information from clinical studies using renal denervation and other neural intervention techniques (e.g. baroreceptor activation). The latter part of the review specifically focuses on pre-clinical data targeting specific cardiac afferents which are known to be sensitized in the setting of chronic heart failure (TRPV1). In response to the Reviewer’s comments, we have combined some sections and clarified the importance and clinical relevance of this review.
Reviewer 2 Report
The author has to clarify why this review is on neuromodulation in CRS-2 even though there are almost no original studies on CRS-2. I do agree that there original studies on neuromodulation in heart failure .It appears that this review appears to be a hypothesis .Place your comments
Author Response
You are correct that we are putting forward an hypothesis that has not been tested clinically. Neuromodulation has been used effectively as indicated to treat drug-resistant hypertension. We have tried to clarify this idea in the revision.
Reviewer 3 Report
This is very important topic. Neuromodulation is very interesting therapy not only for
resistant hypertension and cardiorenal syndrome as described in this review but also in vasovagalsyncope even the target is different.
The paper is well written.
One comment:
The paper includes a lot of important and relevant data: renal denervation, spinal cord stimulation, carotid baroreflex activation, cardiac afferent denervation. I am not sure that the readers will understand what is the clinical implications of neuromodulation in CRS. The authors should add a section (eg. Future direction?) and discuss how they practically suggest to imply neuromodulation in CRS based on the literature and their experience.
I addition, the authors should suggest clinical trials to investigate the clinical significance of neuromodulation in CRS.
Author Response
We completely agree and have added a few sentences in the Conclusions section of the revision.
Round 2
Reviewer 2 Report
Dear author you had been asked to clarify why do you need cardiac denervation CRS-2 but the you have not answered or your answer is not satisfactory
Author Response
I have attached the revised manuscript and the revised figure as recommended by the reviewer.
Reviewer 3 Report
The authors answered my comments
Author Response

(The authors gave the same response as above.)
